# Modeling of High-Resolution Data Converter: Two-Step Pipelined-SAR ADC based on ISDM

**Bo Gao** [1] , **Xin Li** [1] **, Jie Sun** [2] **and Jianhui Wu** [1,*]

[1]  National ASIC System Engineering Center, Southeast University, Nanjing 210096, China;
     gaobo@seu.edu.cn (B.G.); 230179107@seu.edu.cn (X.L.)
[2]  School of electronics and information engineering, Nanjing University of Aeronautics and Astronautics,
     Nanjing 210096, China; dogsun36@163.com
[*]  Correspondence: wjh@seu.edu.cn; Tel.: +86-(025)-8379-3265-8411

**Abstract:** The features of high-resolution and high-bandwidth are in an increasing demand considering to the wide range application fields based on high performance data converters. In this paper, a modeling of high-resolution hybrid analog-to-digital converter (ADC) is proposed to meet those requirements, and a 16-bit two-step pipelined successive approximation register (SAR) analog-to-digital converter (ADC) with first-order continuous-time incremental sigma-delta modulator (ISDM) assisted is presented to verify this modeling. The combination of high-bandwidth two-step pipelined-SAR ADC with low noise ISDM and background comparator offset calibration can achieve higher signal-to-noise ratio (SNR) without sacrificing the speed and plenty of hardware. The usage of a sub-ranging scheme consists of a coarse SAR ADC followed by an fine ISDM, can not only provide better suppression of the noise added in 2nd stage during conversion but also alleviate the demands of comparator's resolution in both stages for a given power budget, compared with a conventional Pipelined-SAR ADC. At 1.2 V/1.8 V supply, 33.3 MS/s and 16 MHz input sinusoidal signal in the 40 nm complementary metal oxide semiconductor (CMOS) process, the post-layout simulation results show that the proposed hybrid ADC achieves a signal-to-noise distortion ratio (SNDR) and a spurious free dynamic range (SFDR) of 86.3 dB and 102.5 dBc respectively with a total power consumption of 19.2 mW.

**Keywords:** Pipelined-SAR ADC; ISDM; noise enhancement; redundant and offset calibration

---

## 1. Introduction

High resolution analog-to-digital converters (ADCs) have been widely used in the medical imaging, radar and high precision industrial control, where it exists a pressing need for power efficient ADCs operating at tens of mega-samples per second (MSPS) [1–3]. The successive approximation register (SAR) ADCs have an advantage of power efficiency because of their simple configuration and digitally oriented character that utilize high-speed and low-power CMOS process [4]. Whereas, the resolution of SAR ADCs is limited within 10 bit because of the thermal noise power (KT/C) noise of sampling network, mismatches of capacitors in the digital-to-analog converter (DAC) arrays and input-referred noise of the comparator [5]. Another shortage of SAR ADC is the limited converting rate owing to the multiple clock cycles necessary for each conversion. Taking the typical 1 bit/cycle architecture for example, one conversion period is approximately equal to $N(t_{comp} + t_{logic} + t_{DAC})$, where $N$ is the total resolution of ADC [6]. Therefore, SAR ADCs are mainly used in low-power and moderate resolution as well as low-speed areas. In the past a few decades, pipeline ADC structures were widely used in high speed and high resolution areas, because their relative stability can be achieved by extensively inter-stage redundancy [7]. However, the feature of high speed and high resolution

is at the expense of power consumption, since for each additional stage of pipeline, one additional high performance gainboost transconductance operational amplifier (OTA) is needed, which means the power consumption will obviously increase [8]. Other structures such as incremental sigma-delta modulator (ISDM) can also realize high resolution by high-order loop filters [9] or high over-sampling radio (OSR) [10], which is time-consuming and is normally adopted in low-power and low speed fields.

Recently, a great number of the hybrid ADCs have been proposed to meet the requirement of high resolution and high speed in the advanced CMOS technology. Pipelined-SAR ADC is one of the typical hybrid-structures, which combines the high-speed merits of pipelining and low-power pros of SAR ADC. In [11], a 12-bit 50 MS/s two-step pipelined-SAR ADC was proposed, which could achieve 66.5 dB signal-to-noise distortion ratio (SNDR) and 78 dB spurious free dynamic range (SFDR) respectively, and expended 3.5 mW only. However, it is difficult to ensure the residue voltage in first stage can stay within in the quantization range of second stage, because the DAC noise, comparator noise and offset in first stage make the residue shift or exceed ±0.5 least significant bit (LSB) possibly with only one inter-stage redundant bit. In [12], a 15-bit 100 MS/s two-step pipelined-SAR ADC was proposed, which used a four-stage ring amplifier (AMP) as the inter-stage gain amplifier, with 73.4 dB SNDR and 90.4 dB SFDR achieved. However, the stability and complexity of ring AMP should be taken into consideration carefully due to the auto-zero in each stage and the employment of high-threshold transistor in the output stage, the similar problem can be found in [13,14]. Therefore, both the stable gain of inter-stage amplifier and enough redundant bits are the key point to design pipelined SAR ADC.

Besides, the resolution of traditional two-step pipelined SAR ADC is limited in many ways. On one hand, the noise of comparator and DAC during SAR conversions cannot be removed because of the open-loop operational SAR logic. To achieve a high SNR, low noise comparator with high power consumption has often been used. However, the trade-off is not generally effective, since suppressing comparator noise for extra one-bit resolution needs four times the power consumption [5]. On the other hand, the distribution of conversion steps between two stages often encounters bottlenecks, which can be described by:

$$\frac{V_{FS,1}}{2^{M-R}} \cdot G = V_{FS,2} = 2^{N-(M-R)} \cdot LSB_2, \tag{1}$$

where $V_{FS,1}$ and $V_{FS,2}$ are the quantization range of 1st and 2nd stage ADC while the G, M (N), and R refer to inter-stage gain, resolution of 1st stage (the whole ADC) and inter-stage redundancy. For one thing, the more bits putted in first stage can get higher resolution, which can release the stress of second stage and be beneficial to the linearity of inter-stage amplifier obviously. However, this means the residue voltage in first stage becomes more sensitive and vulnerable to the noise for many aspects because of the lower residue range. For another thing, the more bits executed in second stage can improve the resolution either, because the noise caused by the 2nd stage can be suppressed evidently by inter-stage amplifier. Whereas it dictates the lower noise and higher power consumption comparator needed in the 2nd stage.

In this paper, a two-step pipelined-SAR ADC is proposed to realize an ADC with the resolution of 16 bit. It combines the SAR ADC and ISDM in the 2nd stage to break current limitation of conventional two-step pipelined ADC by making full use of the timing in 2nd stage. To accelerate the conversion speed in the 1st stage and leave more time for the settling of residue amplification, two DAC arrays including a big one and a small one have been used. Several techniques have been applied to make sure the resolution, linearity and stability of the whole ADC such as the over-range between the 2nd SAR ADC and the ISDM, inter-stage redundancy, back-ground offset voltage calibration of comparator, etc.

This paper is organized as follows. Section 2 describes the proposed architecture of the hybrid ADC and SNR enhancement technique of the ISDM. Section 3 shows circuit implementation details. The pre-layout as well as post-layout simulation results are given in Section 4. Finally, conclusions are drawn in Section 5.

## 2. ADC Architecture and Analysis

### 2.1. The Proposed Architecture Modeling of ADC

As mentioned in Section 1, pipeline ADCs has both merits and demerits. In order to obtain the optimized solution for this design, the trade-offs between resolution and power consumption as well as the number of stages of pipeline ADC were discussed above all. In conventional pipeline ADC, similar bit number of each stage from 2 to 4 bit is widely used based on the 'scaling down' method with 0.5–2-bit inter-stage redundancy to get enough robust characters. Figure 1a illustrates the resolution of pipeline ADC versus the number of pipelined stages. To realize 16-bit resolution, only 2 sub-stages are needed for two-step pipelined SAR, while those pipeline ADCs with the architectures of 1.5 bit/stage, 2.5 bit/stage and 3.5 bit/stage require 11, 7 and 5 stages respectively, which means there are at least the same number of high-performance amplifiers needed in each type. In [8], the total power of pipeline ADC considering to the same noise level for amplifier in each stage is discussed in detail as shown in Figure 1b. With the increase of bit number for each stage, the relative power as well as converting speed of pipeline ADC decreases gradually, which means the more bits put in each stage, the less power consumption will be needed to achieve the similar ADC resolution. In order to obtain adaptable number stage for the features of moderate bandwidth and high-resolution as well as low power, two-step pipelined SAR ADC was used in this design.

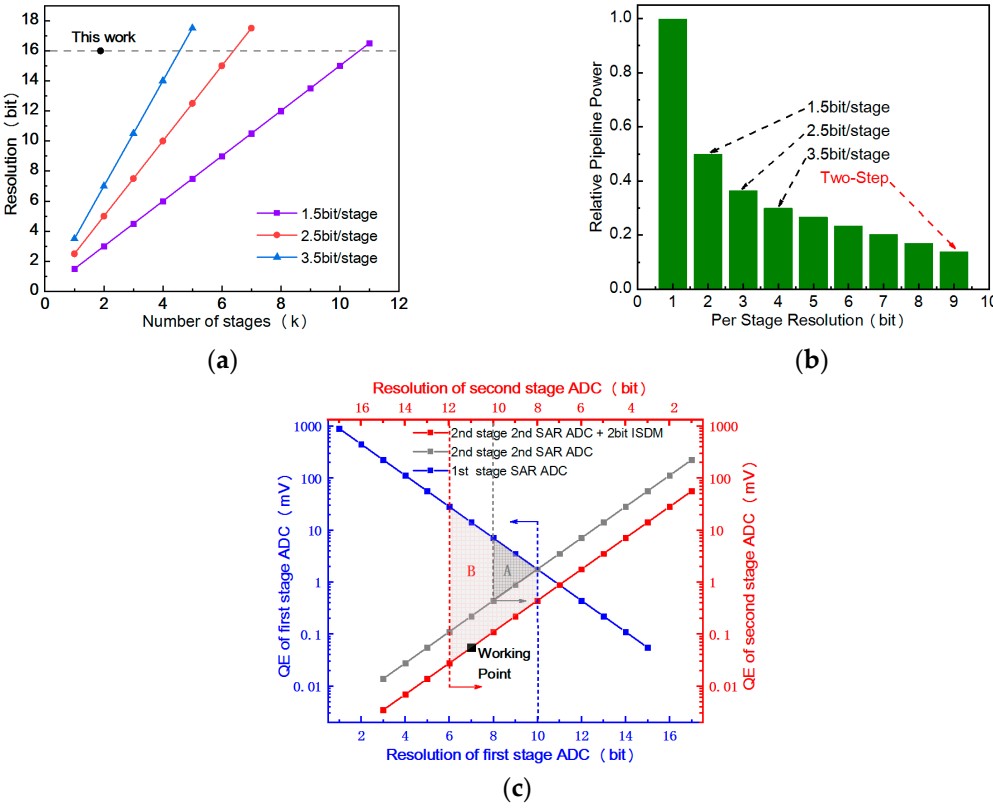

**Figure 1.** (**a**) Resolution versus the number of pipelined stages. (**b**) Relative power versus per stage resolution of ADC. (**c**) Quantization error (QE) of two-step pipelined-successive approximation register analog-to-digital converter (SAR ADC) versus the resolution of first-stage and second-stage SAR ADC w/ or w/o incremental sigma-delta modulator (ISDM) assisted under the Matlab modeling.

In order to determine system-level parameters, Matlab modeling of proposed hybrid ADC structure is carried out. According to formula (1), provided that the $V_{FS,1}$ and $V_{FS,2}$ are equivalent to 1800 mV (differential value), while G and R as well as N are 32 times, 2 bit and 16 bit. Figure 1c shows the equivalent quantization error (QE) of two-step pipelined-SAR ADC versus the resolution of

first-stage SAR ADC and second-stage SAR ADC w/ or w/o ISDM assisted by the Matlab modeling. If assume 10 bit is used as the limit boundary condition of ADC resolution in both stages considering to the input-related noise of comparator without ISDM in the 2nd stage, then the resolution's distributions in the 1st and 2nd stage SAR ADC are (8, 10), (9, 9) and (10, 8), which are within the gray-color regain A in Figure 1. However, even if 8bit is used for the 1st-stage resolution, 32× gain of inter-stage amplifier is not enough to meet the requirements. To break this bottleneck, either doubles the gain to 64× or reduces the reference voltage of the 2nd-stage ADC by half. Both of the solutions are quite challenging, since 64× gain is hard to make sure well linearity in advanced CMOS process and shrink of reference voltage will also cause the extra equivalent gain error between both stages. Hence those three solutions are not recommended to be adopted.

Considering to the assistance of ISDM in the 2nd stage, the resolution of the 2nd stage will be added an extra 2 bit in each case. The distributions of these solutions are within the red-color regain B as shown in Figure 1. Thus, the 1st-stage resolution can be designed below 8 bit. In this paper, the final working point was optimized to (7, 9 + 2), which means that extra 2 bits were obtained by ISDM and the 32× gain without extra gain error calibration needed could be accomplished. Several benefits could be obtained as well. For example, the resolution of the 1st and 2nd stage SAR ADC can be relaxed to only 7 bit and 9 bit, and it will decrease the demands of the comparator's resolution considerably.

Figure 2a shows the proposed two-step pipelined-SAR ADC configuration. Basically, it is composed of a two-step pipelined SAR ADC and a first-order ISDM with binary-weighted DAC arrays in the 1st and 2nd stage. The 1st stage SAR ADC provides 7-bit equivalent resolution and 11 bit in the second stage, with a 2-bit inter-stage redundancy between them. The residue is amplified by the GainBoost AMP 32 times. The DAC arrays in first-stage consist of two parts, the big one and the small one with 7-bit resolution [13]. The 7-bit small DAC array produces binary-shrinking residues as far as possible under the results form coarse comparator of first stage with the asynchronous timing control logic. While the 7-bit big DAC takes charge of generating final residue of first stage with the 7-bit binary codes obtained from the shift register in small DAC as soon as finishing conversion of $LSB_1$. The total sampling capacitance of DACs is about 8.7 pF, which can achieve a near 14-bit KT/C noise level. Nonetheless, the small DAC needs a KT/C noise level up to 7-bit only. In this case the input-referred noise requirement of the comparator in 1st stage is greatly relaxed, as well as the power of it.

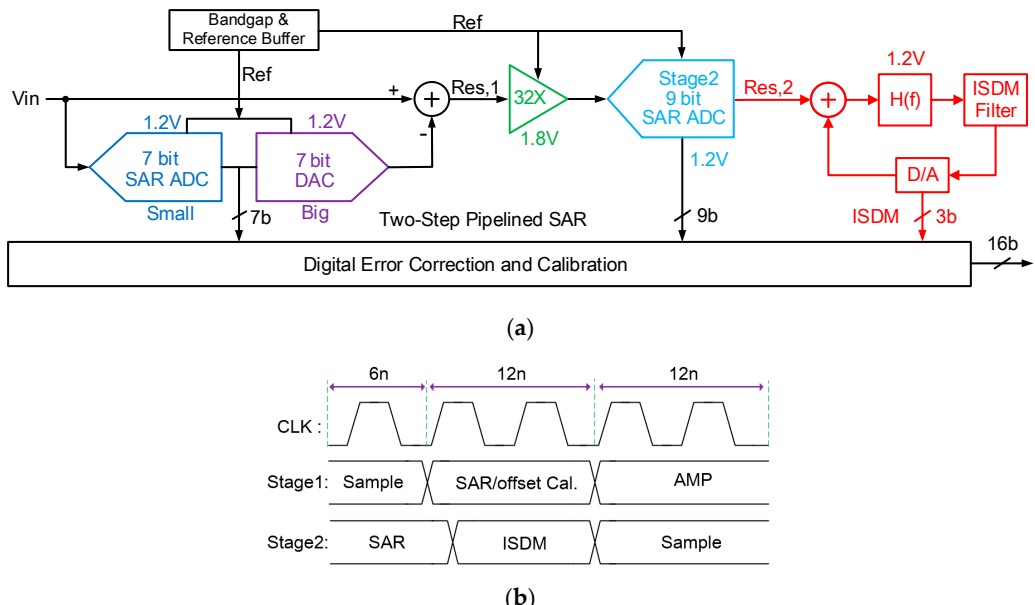

(a)

(b)

**Figure 2.** (**a**) The architecture of proposed two-step pipelined-SAR ADC with ISDM assisted. (**b**) The timing distribution of the whole pipelined SAR ADC.

The 2nd stage consists of 9-bit SAR ADC and 3-bit ISDM with 1-bit over-range between them to get the final 11-bit equivalent resolution n, the 1 bit over-range added between the coarse SAR ADC and the ISDM in order to tolerate the coarse SAR ADC error induced by the SAR loop thermal noise. The ISDM loop converts the residue of coarse SAR ADC in the 2nd stage for eight times in eight synchronous cycles generated by a delay-locked loop (DLL) clock generator. Those eight comparing decisions of the ISDM are filtered and decimated using a FIR mean filter by eight times, getting the final 3-bit binary codes. Besides, the reference buffer provides the same reference voltage for DAC arrays in the 1st and 2nd stage, $V_{RP} = 1.05$ V and $V_{RN} = 0.15$ V from an on-chip bandgap. Digital error correction and calibration (DEC) logic combines the binary codes from first and second stage and outputs 16-bit binary codes finally.

The timing distribution of the whole pipelined SAR ADC is shown in Figure 2b. The input signal is sampled using bottom-plate of DAC arrays by a sort of optimized bootstrap switch [15] in 6 ns, and then the first stage begins to convert the sampled signal in 12 ns with the $V_{CM}$-based switching scheme of big-and-small DAC arrays. The background calibration for comparator's offset of first stage is executed during this period. Afterwards, the residue voltage is amplified by the GainBoost AMP by 32 times in 12 ns. Meanwhile, the sampling of second stage hybrid SAR-ISDM ADC is carrying out with the bottom-plate of DAC array as well. After that the coarse conversions of second stage SAR ADC will begin with the control of asynchronous self-time clock generated by the latch in second stage comparator, with the ISDM fine conversions followed, both of the processes consume totally of 18 ns.

### 2.2. SNR Enhancement Technology

In this paper, the continuous-time ISDM inserted in the 2nd stage breaks those bottlenecks as mentioned in Section 1 [16]. With only 7 bit and 9 bit SAR conversions in the 1st and 2nd stage respectively, the demands for comparators in both stages can be decreased remarkably. Meanwhile, the close-loop characters of ISDM with 8× OSR make it possible to achieve higher resolution without too much extra power penalty.

Figure 3a shows the block diagram of ISDM in the second stage. $V_A$ is different between $V_{RES,2}$ and $V_{SD}$, where $V_{RES,2}$ and $V_{SD}$ stand for the quantization error of the 2nd stage SAR ADC and reference voltage for 1 bit DAC of the ISDM respectively. In this design, the $V_{SD}$ equaled 2 LSBs of SAR ADC so that the $V_{RES,2}$ could be within the quantization range of ISDM, considering the noise existed on the top-plate of DAC practically. $V_{INT}$ is the output of Integrator after $V_A$ integrated by the continuous-time integrator eight times. Besides, the open-loop integrator has been used in this paper to save power as much as possible comparing to the traditional close-loop and power-hungry integrator [17].

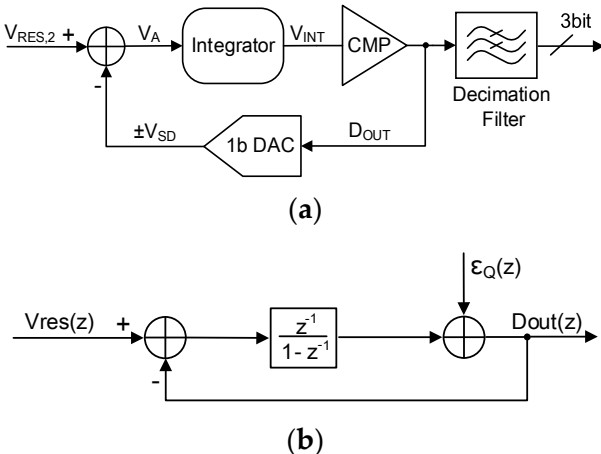

**Figure 3.** (**a**) The proposed signal flow diagram of first-order continuous-time incremental sigma-delta modulator. (**b**) First-order z-transform domain model of it.

The first-order z-transform domain model of this continuous-time ISDM is shown in Figure 3b, where $\varepsilon_Q$ is the noise in ISDM close loop, including the quantization noise, comparator thermal-noise and loop filter noise. If the gain of integrator can keep constant and then the output of ISDM ($D_{OUT}$) is written as:

$$[V_{RES,2}(z) - D_{OUT}(z)]\cdot\frac{z^{-1}}{1-z^{-1}} + \varepsilon_Q = D_{OUT}(z). \tag{2}$$

Based on (1) and the signal flow diagram, the transfer function of the ISDM can be obtained as:

$$D_{OUT}(z) = V_{RES,2}(z)\cdot z^{-1} + \varepsilon_Q(z)\cdot(1 - z^{-1}), \tag{3}$$

where $\varepsilon_Q$ is first-order shaped by the noise transfer function (NTF): $1 - z^{-1}$, with the input $V_{RES,2}$ passes through a delay $z^{-1}$ only [18]. In this paper, the $V_{RES,2}$ was integrated in eight times while it kept in no change, which was similar to the case that the $V_{RES,2}$ was integrated in only one time by a sampling frequency of 8·fs and kept the Nyquist signal bandwidth (f$_B$) at the same time (see the illustration in the Figure 4a), and it means the OSR was 8. Therefore, the noise power caused by $\varepsilon_Q$ was shifted beyond f$_B$ significantly, and then the noise would be low-pass filtered by the FIR mean filter. Figure 4b shows the bode-diagram of the NTF. Due to an OSR of eight times, the SNR of 2nd stage ISDM-SAR ADC would be improved more than 12 dB within eight times continuous-time integrating options (with one-bit over-range) in theory.

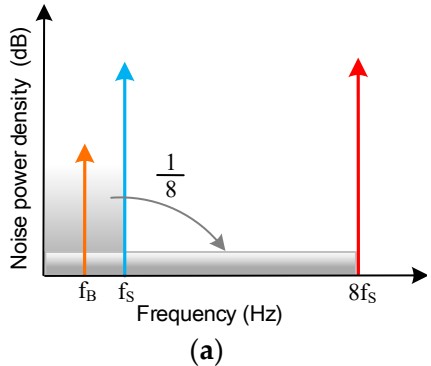 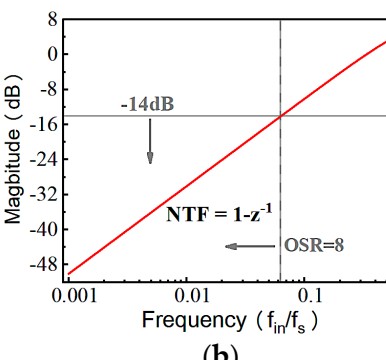

(**a**)　　　　　　　　　　　　　　　　　　　　　　(**b**)

**Figure 4.** (**a**) The noise power density distribution. (**b**) The bode-diagram of the noise transfer function of the ISDM.

## 3. Circuit-Level Implementation

### 3.1. Inter-Stage Amplifier

As illustrated in Section 2.1, the residue of first stage needs to be amplified 32 times by inter-stage gain amplifier. There are mainly about three types of amplifier that can be used in pipelined-SAR ADC, including the dynamic amplifier, ring amplifier and GainBoost amplifier. However, due to the instability, limited gain and large gain error, the dynamic amplifier can be just used in middle resolution pipelined-SAR, excepting for those designing with complexes calibration [19]. Meanwhile, the utilization of ring AMP [13] in high-resolution pipelined-SAR ADC has to face some certain problems. For example, the open-loop gain is limited and hardly even exceeds 90 dB and the considerably large setting time makes it difficult to use in high speed applications. Comparing to other two kinds of amplifiers, the GainBoost OTA has significant merits such as high open-loop gain and fast setting speed, which has been widely applied in pipelined ADC.

In this design, the two-stage GainBoost OTA was used as the inter-stage amplifier. As is shown in Figure 5, the first stage is a conventional telescope GainBoost structure [20]. In order to decrease the power as much as possible, the 1st and 2nd stages of SAR ADC use low voltage (1.05 V and 150 mV) as the reference voltage whereas the traditional GainBoost OTA needs at least 1.8 V supply voltage,

which means the output common mode voltage of GainBoost needs to meet the demands of the input common mode voltage of the 2nd SAR ADC. To remedy this problem, the common-source amplifier with positive channel Metal Oxide Semiconductor (PMOS) input pair was used as the second stage of GainBoost OTA. In this way the output common could be set to 600 mV stably with the switch-capacitor common feedback. Besides, the thick oxide PMOS was used near the high supply voltage (VDDH), which decreases the parasitic capacitance and power considerably [21].

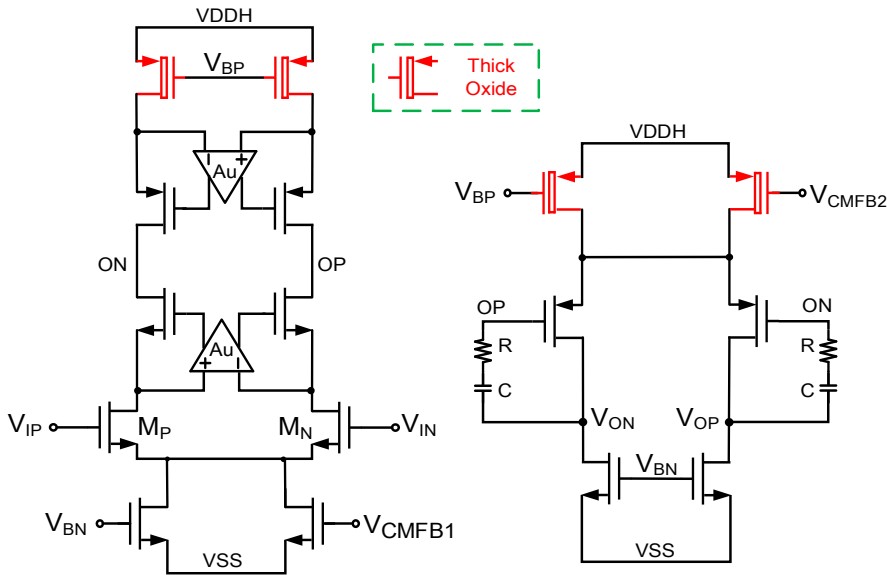

**Figure 5.** The circuits of proposed two-stage GainBoost OTA.

Assuming that the residue voltage of the 1st stage ADC has enough time to settle and the static settling error ($\varepsilon_S$) is about half of quantization error ($\varepsilon_Q$) of second stage ADC, then the output of amplifier can be expressed as:

$$V_o = V_{in} \cdot \frac{1}{1 + \frac{1}{A_0 \cdot \beta}} \approx V_{in} \cdot (1 - \frac{1}{A_0 \cdot \beta}), \tag{4}$$

$$\varepsilon_S = \frac{1}{A_0 \cdot \beta}, \tag{5}$$

where $A_0$ and $\beta$ are open-loop gain and loop index of GainBoost OTA respectively. As shown in (4), the static error is inversely proportional to loop gain ($A_0 \beta$). In order to satisfy the resolution demand of the 2nd stage SAR ADC, the loop gain of GainBoost is limited by the quantization error ($\varepsilon_Q$) of second ADC:

$$A_0 \cdot \beta > 2^{N+1}. \tag{6}$$

Since the resolution of second stage is 11 bit, the loop gain is at least 72.2 dB. Simulation results of the loop stability of GainBoost OTA is shown in Table 1, where the low-frequency loop gain and unity-gain bandwidth (GBW) were 79.3 dB and 328.3 MHz respectively, with the phase margin beyond 70°. To avoid the phenomenon of zero-pole doublet caused by applying the auxiliary amplifier [22], the main pole of which ($P_{AU}$) must meets the formula:

$$\beta \cdot GBW < P_{AU} < P_2, \tag{7}$$

where, $P_2$ is second pole of main amplifier in the 1st stage of GainBoost OTA. In fact, Equation (7) is easy to achieve by adding a small capacitance at the output of the auxiliary amplifier.

In Table 1, C and R stand for the Miller compensation's networks between the 1st and 2nd stages of GainBoost OTA, which contributed to the phase margin in 73.5° finally. In order to decrease the slew time of OTA as much as possible and leave more time for linear settling. Assuming that the slew time was a tenth of the total amplifying time (12 ns), then the current in second stage of OTA was set to 1.46 mA, which made the final slew time near 1.1 ns with a slew rate of about 560 V/μs. The transient simulation proves the proposed GainBoost OTA can achieve a resolution of 12.4 bit ultimately. Due to the more than 100 dB open-loop gain, simulation result shows that the input-referred noise ($V_{n,OTA}$) of the GainBoost OTA was only about 27 nVrms, which is small enough comparing to 1 LSB (7 mV, single end) of the 1st stage SAR ADC and was thus ignored totally. Meanwhile, the considerable sizes of input-pair transistors (MP and MN as shown in Figure 5) made a remarkable matching with a little offset ($V_{offset,\,OTA}$ = 183 μVrms) between them, which could be tolerated by the inter-stage redundancy.

**Table 1.** The optimized parameters of Gainboost OTA.

| Parameter | Value | Parameter | Value |
|---|---|---|---|
| Loop Gain (dB) | 79.3 | Phase Margin (°) | 73.5 |
| GBW (MHz) | 328.3 | $A_0$ (dB) | 104 |
| $1/\beta$ | 32.15 | P2 (GHz) | 1.87 |
| $P_{AU\_P}$ (MHz) | 704 | $P_{AU\_N}$ (MHz) | 755 |
| $C_S$ (pF) | 4.096 | $C_L$ (pF) | 2.05 |
| C (fF) | 126 | R (kΩ) | 1.2 |
| Slew Rate (V/μs) | 560 | $V_{n,OTA}$ (nVrms) | 27 |
| SNDR (dB) | 75 | SFDR (dBc) | 84 |

### 3.2. First Stage SAR ADC

The structure of first stage SAR ADC is shown in Figure 6a. It mainly consists of DACs, sampling switches, comparator and control logic. As illustrated in Section 2.1, the first stage SAR ADC has two parts of small and big DAC arrays. The unit capacitances of them ($C_{1S}$ and $C_{1B}$) are 4.86 fF and 29.14 fF respectively, with a ratio of 1:6 between them. Those two DACs sample the input signal by the same bootstrap circuit whereas the switching transistors for each bottom plate of capacitor keep in different size to get the similar RC constant. The timing of the whole ADC is shown in Figure 6b. After the sampling phase (CKS = 6 ns, CKSE is the early shutdown clock), the SAR conversions will be carried out within seven cycles, which is generated by asynchronous self-time clock logic within the enable clock (SAR_EN). Once the LSB of 1st SAR ADC ends, 7 bit binary code from register will control the reference voltage to settle the final residue for big DAC with $V_{CM}$-based switching algorithm [23]. In this design, two kinds of common-mode voltage were utilized, including $V_{CMS}$ (600 mV) for small DAC and $V_{CMA}$ (750 mV) for the top plate of big DAC, which could match the input common-voltage of amplifier.

Figure 7 illustrates the sampling networks model of the 1st stage DAC arrays in Figure 6a. According to the charge conservation, the signal on the top plate of DAC after sampling can be expressed by:

$$V_S = V_{CMS} + \frac{C_{S1}}{C_{S1}+C_{S2}}(V_{CMS} - V_{in}), \tag{8}$$

$$V_B = V_{CMA} + \frac{C_{S1}}{C_{S1}+C_{S2}}(V_{CMS} - V_{in}), \tag{9}$$

where, $V_S$ and $V_B$ are the sampled signals on the top plate of small and big DAC arrays respectively as can be seen from the formulas (8) and (9). Input differential signal will be attenuated by different coefficients ($C_{S1}/(C_{S1} + C_{P1})$ for small DAC and $C_{S2}/(C_{S2} + C_{P2})$ for big DAC), of which $C_{S1}$, $C_{S2}$ and $C_{P1}$, $C_{P2}$ are the total sampling capacitance and parasitic capacitance of the top plate in small and big DAC arrays respectively.

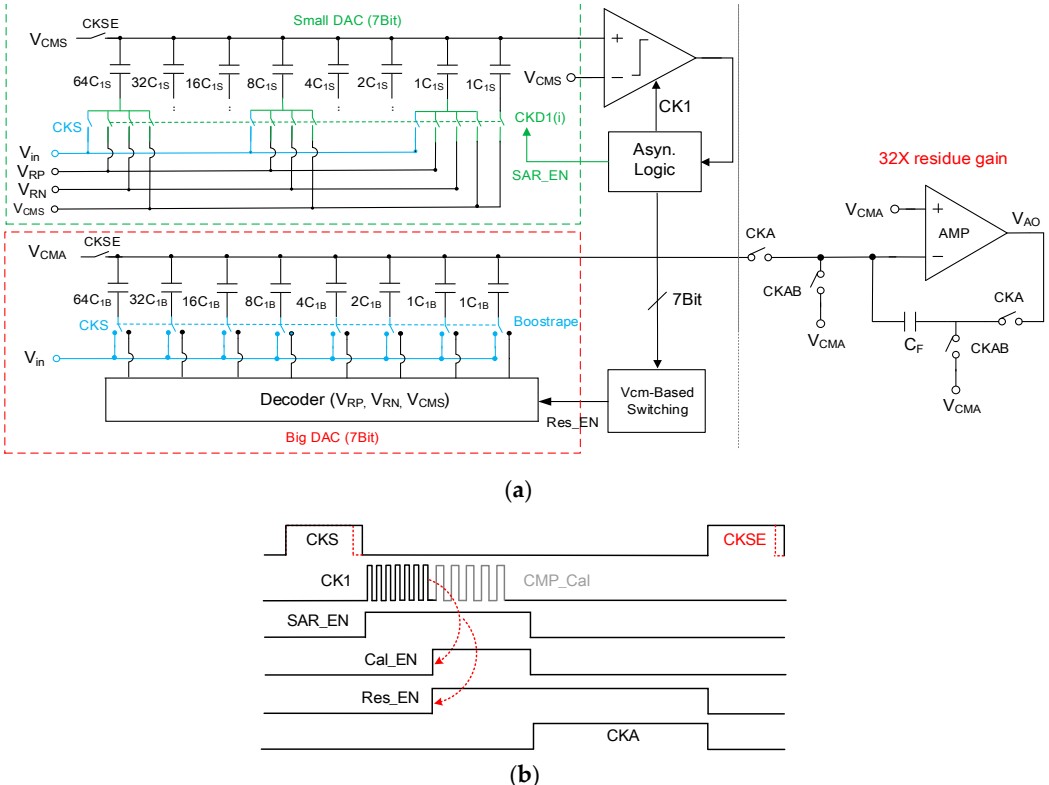

(**a**)

(**b**)

**Figure 6.** (**a**) The diagram of 1st stage SAR ADC and inter-stage gain amplifier (actual implementation is fully differential. (**b**) The operation sequence of them.

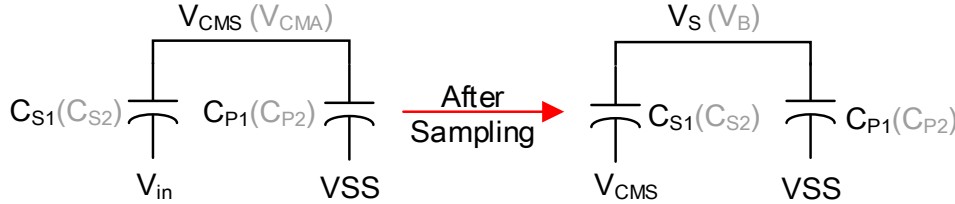

**Figure 7.** The sampling networks model of 1st stage DAC.

### 3.3. Second Stage ISDM SAR ADC

The 2nd stage ADC consists of 9 bit SAR ADC and 3 bit ISDM, including two main processes: The SAR conversion and the ISDM quantization. The unit capacitance of DAC ($C_2$) was 8 fF with a total capacitance of 2.05 pF, which could meet the demand of the KT/C noise limitation of 11 bit. Figure 8a shows its diagram and the timing is shown in Figure 8b. When the residue of the 1st stage was amplified by the inter-stage amplifier during the high level voltage period of CKA (12 ns), the output of amplifier ($V_{AO}$) was sampled by the 2nd DAC within CKS2 (CKSE2 is the early shutdown clock) at the same time. After sampling, the SAR conversions would be underway in nine cycles, which was generated by asynchronous self-time clock logic within the enable clock (SAR_EN2). Once the LSB of the 2nd SAR ADC (LSB$_2$) ends, the residue voltage ($V_{RES,2}$) on the top plate of DAC keeps constant and the ISDM will begin to execute within ISDM_EN phase. Meanwhile, the FIR mean filter begins to count how many times the output result of comparator is '1' by a 3-bit binary counter as shown in Figure 8. After the decimation those 3-bit original output code (SDo<0:2>) combines with the output of 2nd SAR ADC and makes the final 11 bit output code of 2nd stage ADC. The clock for ISDM is eight synchronous cycles generated by DLL, which can make sure the integration time for each cycle is equal.

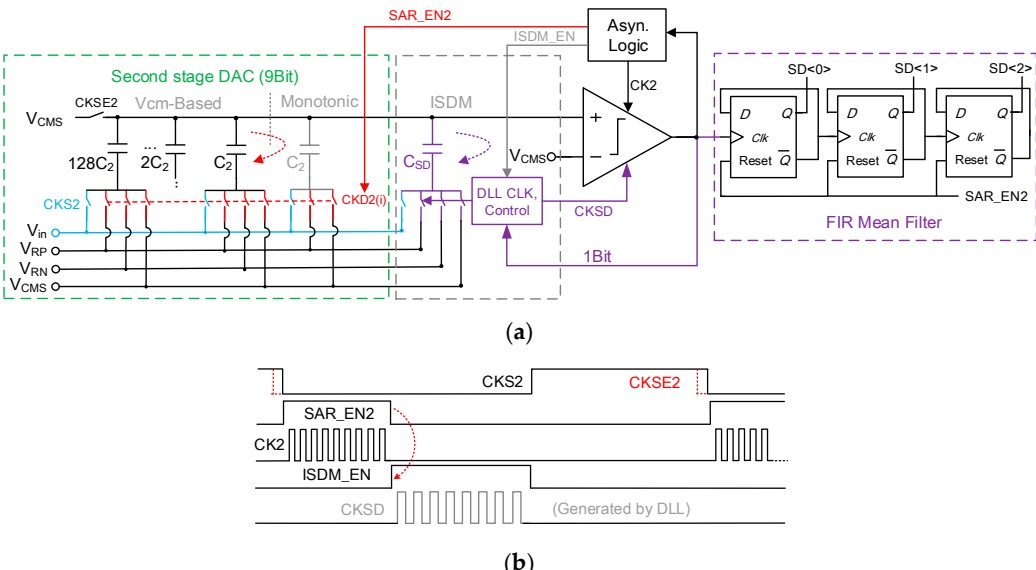

**Figure 8.** (**a**) The diagram of the 2nd stage ISDM SAR ADC (actually implementation is fully differential. (**b**) The operation sequence of it.

As mentioned in Section 2.1, 1-bit over-range was added between the SAR ADC and ISDM, which means the capacitance of DAC for ISDM ($C_{SD}$) was two times of $C_2$. However, the DAC of 2nd SAR ADC was 8 bit with the same unit capacitor ($C_2$ = 8fF) for LSB and LSB + 1. In this paper, LSB adopted the monotonic switching scheme [24] whereas conversions from LSB + 1 to MSB adopted $V_{CM}$-based switching scheme. In this way, the common voltage on top plate of ADC kept near 600 mV whereas the unit capacitance was doubled compared to conventional 9-bit SAR ADC, which could relatively decrease mismatch of capacitance in DAC. Therefore, the $C_{SD}$ was equal to $C_2$ actually. Assuming that the gain of integrator ($A_{INT}$) for ISDM is constant after the integration of ISDM, the output voltage of integrator ($V_{INT}$) can be expressed by:

$$V_{INT} = \left[ V_{RES,2} - \sum_{i=1}^{M} V_{SD}(i) \right] \cdot A_{INT} + \varepsilon_Q, \tag{10}$$

where $V_{SD}$ is the maximum quantization range of ISDM generated by $C_{SD}$, which is near 2 LSBs of the 2nd stage SAR ADC. $M$ is the total number of '1' of comparing results after eight times the integrations and the $\varepsilon_Q$ stands for the quantization error as mentioned in Section 2.2.

### 3.4. Comparator and Integrator

Schematic of the comparator in the 1st stage is shown in Figure 9, and it consists of a pre-amplifier stage and a dynamic latch stage [25]. In this paper, the total sampling capacitance of small DAC was about 622 fF, which indicates the parasitic capacitance ($C_{P1}$) result in the attenuation on the sampled signal. In order to decrease the influence of parasitic from input-pair devices as much as possible, the sizes of M2 and M3 need to be optimized carefully whereas it also had to balance the tradeoff between the performances of the resolving time, noise and offset. In this design, the M4 and M5 were cascaded upon the drain of M2 and M3 respectively while the gates of M6 and M7 were connected in a cross-coupled way, which could increase the impedance of VP and VN properly and decline kick-back noise from dynamic latch either. The dynamic latch stage adopted PMOS as an input-pair instead of connecting the input to the cross-coupled nodes of latch directly, which helped reduce kick-back noise to the top plate of DAC further [26].

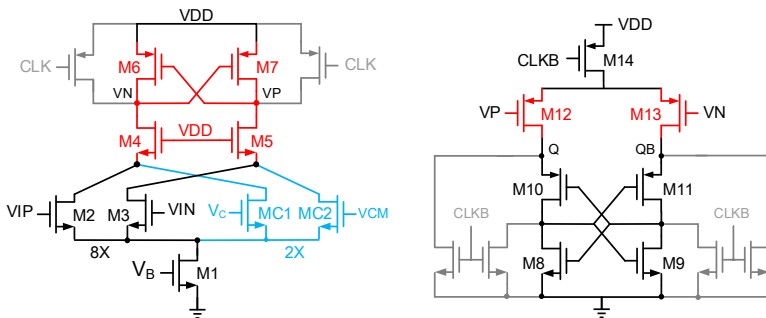

**Figure 9.** The diagram of comparator in the 1st stage SAR ADC.

The comparator used in the 2nd SAR ADC is shown in Figure 10a, which includes an integrator stage and a comparator with the same structure of that in the 1st SAR ADC, whereas the sizes of input-pair were bigger than that in the 1st comparator to meet the higher resolution (9-bit) demands of the 2nd SAR ADC. In this design, the PMOS differential pairs (M3 and M4) were used to decrease the input-referred noise during the integrating phase as much as possible while the NMOS M1 and M2 were connected in a cross-coupled way to get well gain income. The trail current of differential pairs was determined by current source M6. It can provide a stable DC current, then the gain of integrator ($A_{INT}$) can be set in a near constant state with a small input signal (within 5 mV) during the ISDM process, which can be expressed by:

$$A_{INT} = \frac{g_m \cdot \tau}{C_{INT}}, \tag{11}$$

where $g_m$ and $\tau$ are transconductance of differential pairs and integrating time. $C_{INT}$ is the integrating capacitor between the differential output nodes of the integrator, which plays a role of the load when the SAR conversion and reset by RST (see the illustration in the Figure 10b) once the end of each comparing cycles whereas keeping without the reset during the period of continue-time integrating. Due to the implementation of $V_{CM}$-based and monotonic switching schemes in 2nd SAR ADC, the input common of comparator is near $V_{CMS}$, which mean the $g_m$ can keep in constant with a small input amplitude. Besides, the DLL helps generate relatively stable CLK for the ISDM phase, which means the $\tau$ is constant either. In this way, the $A_{INT}$ is stable to meet the demands of integrating loop.

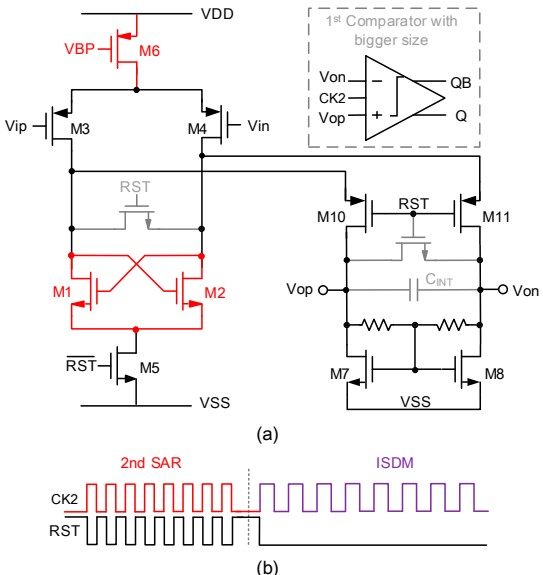

**Figure 10.** (**a**) Diagram of integrator based comparator in 2nd stage SAR ADC. (**b**) The working sequence of it.

### 3.5. System Stabilization and Calibration

The key point to keep the system stabilization of pipelined-SAR ADC includes two aspects. The first one is to make sure the amplified residue from the 1st stage SAR ADC is within the quantization range of the 2nd stage ADC (see the explanation in Figure 11a). The second is to ensure the linearity and resolution of inter-stage amplifier meets the demands of 2nd stage ADC. As is depicted in Section 3.1, the amplifier was designed to satisfy design requirements with 75 dB SNDR and 84 dB SFDR respectively, leaving the first point to be resolved. After the conversion of the 1st stage ADC, the residue ($V_{\text{res},1}$) can be expressed:

$$V_{\text{res},1} \approx \varepsilon_1 + V_{mis} + V_{offset,cmp} + V_{n,cmp} + V_{n,DAC} + V_{n,\text{OTA}}, \tag{12}$$

where the $\varepsilon_1$ is quantization error of the 1st stage SAR ADC while $V_{offset,cmp}$, $V_{n,cmp}$ and $V_{n,DAC}$ stand for the offset voltage, input-referred noise of comparator and the thermal noise of DAC in the 1st stage. The offset caused by comparator ($V_{offset,cmp}$) and capacitor mismatch ($V_{mis}$) between small and big DAC in the 1st stage SAR ADC needs serious considerations, since the output binary code of ADC will appear as missing codes once the amplified-residue from 1st stage ADC exceeds the quantitation range of 2nd ADC, which may lead to the breakdown of the whole ADC (see the explanation in Figure 11b). As is mentioned in formulas (8) and (9), the parasitic capacitances ($C_{P1}$ and $C_{P2}$) on the top plate of DAC arrays are different between small and big DAC, which means the residue settled by big DAC has a gap comparing to small DAC. In order to solve those problems, 2-bit inter-stage redundancy was applied between the 1st and 2nd stage ADC to tolerant the offset caused by DAC arrays and input-referred noise of the 1st comparator whereas its offset needs to be calibrated alone.

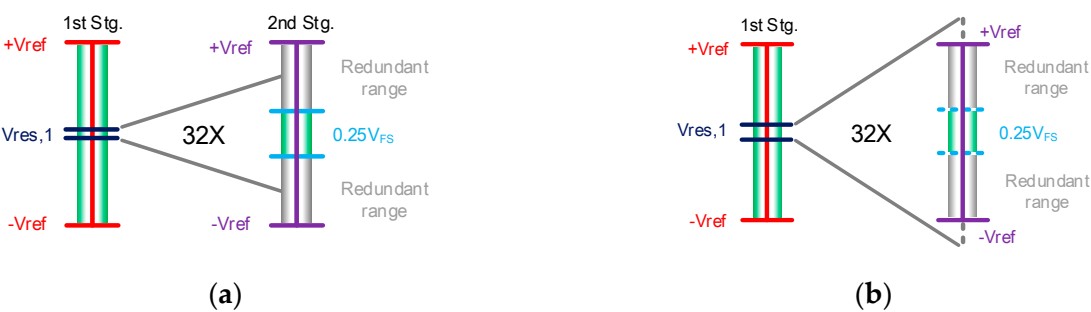

**(a)**                                                                                                          **(b)**

**Figure 11.** The diagram of residue transforming from 1st stage to 2nd stage: (**a**) the residues are in the redundant range; (**b**) the residues are out of the redundant range.

In order to estimate the input-referred noise and offset voltage of comparator located in 1st stage ADC, the methods proposed in [27] adopted to calculate the $V_{n,cmp}$ and the result is shown in Figure 12a. With 16 equally spaced input signals from $-\text{LSB}_1$ to $+\text{LSB}_1$, the fitting curve shows the standard deviation ($\sigma_{n,cmp}$) of $V_{n,cmp}$ was near 1.4 mV ($0.1\text{LSB}_1$, the LSB of the 1st stage). Due to the randomness of $V_{n,cmp}$, it can only be tolerated by inter-stage redundant. Meanwhile the Monte Carlo simulation was carried out by 1000 times to evaluate the $V_{offset,cmp}$ and the results as shown in Figure 12b. The root mean square value of the offset voltage was 347 μV whereas its $3\sigma$ was 17.4 mV, which exceeded $1\text{LSB}_1$ (14 mV). Therefore, the input-referred offset would affect the resolution of 1st stage directly if the mismatch of devices was considerable. In this paper, the scheme for the calibration of comparator is shown in Figure 13. It was similar to [28], whereas the calibration reference voltage gained was quietly different. The resistance string was chosen rather than calibration capacitor unit because it is hardly ever possible to make sure the charge and discharge time is equal, which probably leads to the accumulation of charge on the calibrating capacitor and then a big shift of the common voltage of calibration transistor may influence the calibrating resolution in practice.

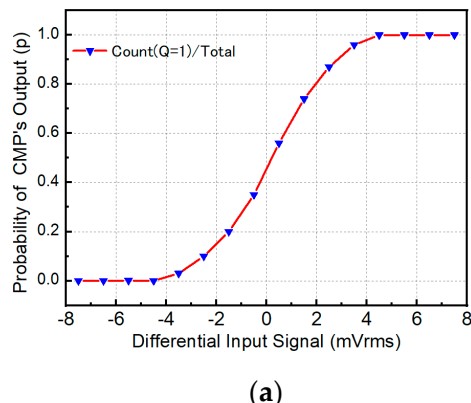

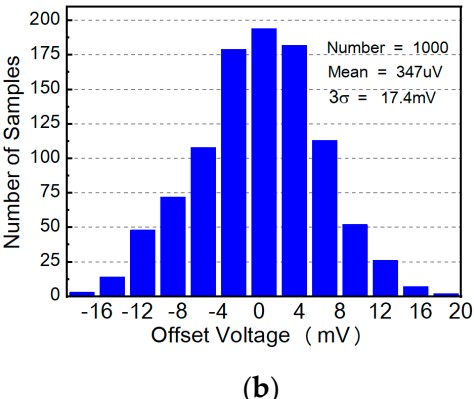

(**a**)    (**b**)

**Figure 12.** (**a**) Fitting of input-referred noise ($\sigma_{n,\text{CMP}}$) of the comparator. (**b**) The Monte Carlo simulation of offset voltage in 1st stage SAR ADC.

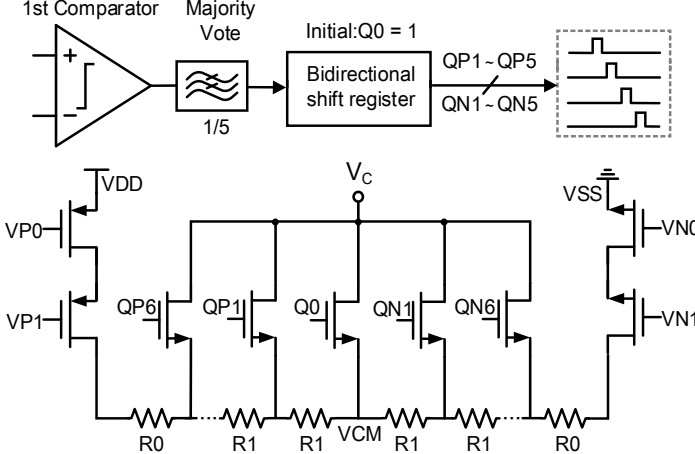

**Figure 13.** The modified scheme of the calibration of comparator in the 1st stage SAR ADC.

After the conversion of the LSB in the 1st stage, the top plate of small DAC array would be shorted to $V_{CMS}$ while the gates of the calibration transistor $M_{C1}$ and $M_{C2}$ were connected to VCM generated by resistance string, which was biased by the current source to isolate noise from the supply voltage as much as possible. The size of $M_{C1}$ and $M_{C2}$ is a forth of input-pair M2 and M3, which can relax the resolution demand of resistance string and decrease the kick-back noise properly. Then the calibration process will begin within five asynchronous cycles (see the illustration in Figure 8b). As is illustrated in Figure 13, the output of the majority voting block will control the bidirectional shift register to generate the calibration control signal (QN1-N6 for positive offset voltage and Q0 for initial voltage with QP1-P6 for negative offset voltage). In order to cover at least $2\sigma$ (12 mV) offset, the resistance string had 12 stairs to generate relative reference voltages with a near 4.7 mV (because 4× gain was put between input pair and calibration transistors as shown in Figure 9) step between them.

In this paper, the concept of the majority voting scheme [29] was adopted to obtain the final calibration direction after five times of the comparisons. It can contribute to decreasing the influence of input-referred noise of comparator remarkably. If there are five times (counting by a 3-bit counter) the comparison results are '1', which means the offset voltage is positive, then the output of the majority voting block is 'Down' and calibration voltage ($V_C$) will drop one step (see the illustration in Figure 14). If there are five times the comparison results are '0', which means the offset voltage is negative, then the output of the majority voting block is 'Up' and $V_C$ will rise one step. In addition, the times of '1' between 2 and 4 means the output are 'Hold' and the $V_C$ keeps without any action.

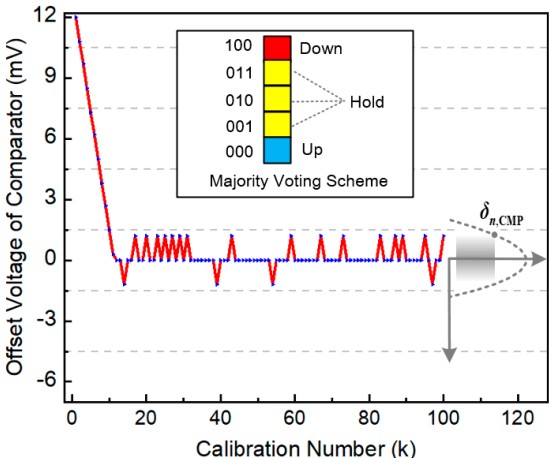

**Figure 14.** The convergence curve of input-referred offset voltage ($V_{offset,cmp}$) of the 1st comparator and the majority voting scheme.

Figure 14 shows the simulation result of the proposed offset calibration of the 1st comparator. With 12 mV ($2\sigma$) offset voltage added on the input port of comparator, the curve shows that the $V_{offset,cmp}$ would converge in ±1.5 mV (it is comparator threshold) within 12 times of calibration process. In this way, the $V_{offset,cmp}$ and $V_{n,cmp}$ were both near $0.1LSB_1$, while the $V_{n,DAC}$ could be ignored because of the relatively big sampling capacitance as mentioned in Section 2.1. Besides, the input-referred noise ($V_{n,OTA}$) and offset ($V_{offset,OTA}$) of the inter-stage amplifier were both smaller (see the details in Section 3.1) compared to those of the comparator. Assume all of the non-ideal factors can be expressed by the $V_{ND}$, then:

$$V_{ND} \approx V_{mis} + V_{offset,cmp} + V_{n,cmp} + V_{n,DAC} + V_{offset,OTA} + V_{n,OTA}. \tag{13}$$

Combine formulas (10) and (11), and the $V_{res,1}$ can be expressed.

$$V_{res,1} = \varepsilon_1 + V_{ND}. \tag{14}$$

The simulation results show that the $V_{mis}$ caused by capacitor mismatch between small and big DAC in the 1st stage SAR ADC was about 0.36 LSB, then the whole non-ideal factors ($V_{ND}$) in formulas (13) and (14) was less than 0.6 LSB, which would take 0.6 bit inter-stage redundancy and leave more than one redundant bit to tolerant other non-ideal factors. Therefore, the whole two-step pipelined-SAR with ISDM assisted could work in a stable state and realize the final output of the 16 bit digital code.

## 4. Simulation Results

The prototype 16-bit two-step pipelined-SAR ADC was implemented in the 40 nm CMOS process and it operated at 33.3 MS/s under 1.2 V/1.8 V supply with the total power of 19.2 mW, including the on-chip background calibration. Figure 15 shows pre-layout simulation results about SNDR and SFDR with a 2.3 MHz input signal. Comparing to the conventional 14 bit pipelined-SAR ADC, the proposed 16 bit hybrid two-step pipelined SAR ADC with ISDM assisted could achieve an SNDR of 96.74 dB and SFDR of 115.30 dBc respectively, with a remarkable improvement of SNDR near 12 dB. Figure 16a shows the simulated SNDR and SFDR versus input signal frequency from low frequency to Nyquist with different process corners (TT, SF and FS). SNDR was better than 91 dB in the first Nyquist zone with the SFDR beyond 103 dBc. Therefore, this kind of design shows remarkable robust characteristics to the process variation. The effective number of bits (ENOB) distribution of ADC versus the number of Monte-Carlo run with capacitor mismatch is shown in Figure 16b, comparing with 4% of the capacitor

mismatch added on DAC arrays, 2% of mismatch caused only about a 0.2-bit ENOB drop, which was due to the relatively enough redundant bits between the 1st and 2nd stage of ADC, hence the demand of capacitor mismatch capacitors could be relaxed.

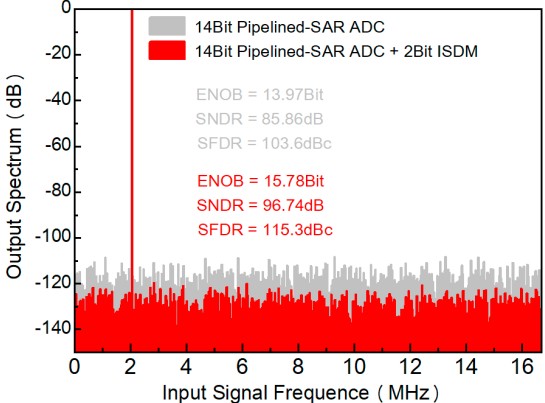

**Figure 15.** The output spectrum of the 16 bit pipelined-SAR ADC (when Fin = 2.3 MHz and Fs = 33.3 MHz) under the pre-layout simulation.

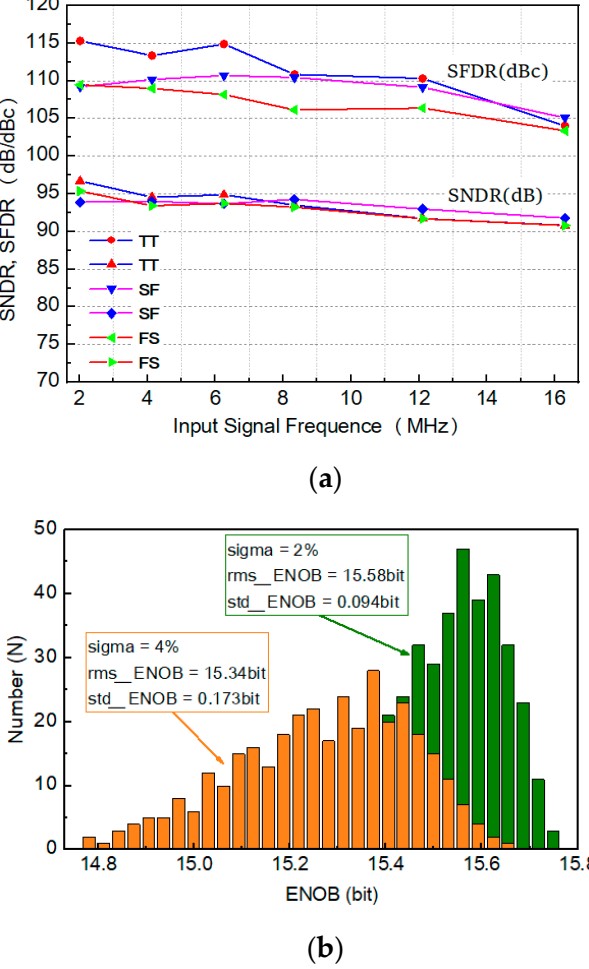

**Figure 16.** (**a**) The pre-layout simulation about the dynamic performance of the whole pipelined-SAR ISDM ADC versus input frequency with different process corners. (**b**) ENOB distribution versus number of Monte-Carlo run with capacitor mismatch ($\sigma$ = 2% and $\sigma$ = 4%) added on DAC arrays as well as non-ideal factors as mentioned in (13) under Matlab simulation.

Figure 17 shows the layout photography of the proposed 16-bit pipelined-SAR ADC with ISDM assisted, which mainly consists of the 1st and 2nd stage SAR ADC and inter-stage amplifier, band gap, input and reference buffer with a core area of 0.202 mm². The post-layout simulation results of SNDR versus Monte Carlo (MC) are shown in Figure 18. With a 2.3 MHz input signal, the SNDR was around 86.3 dB. Comparing to the pre-layout simulation results, the SNDR dropped nearby to 10.4 dB, which was mainly caused by added transient noise and mismatch. Other reasons like the interconnected parasitic capacitor and inductance of the I/O port as well as supply ripples (1%, an extreme case) could also lead to the drop of the performance. Figure 19 shows representative FFT spectra taken at a 16 MHz and −0.4 dBFS input sinusoid signal under post-layout simulation with transient noise added to verify the design and compare with other works. The proposed ADC could achieve a high performance with an ENOB exceed 14-bit. Meanwhile, the SNDR and SFDR were more than 86 dB and 102 dBc respectively with THD near 98.2 dB. Anyway, the post-layout simulation had shown a properly superior performance comparing to some recent Nyquist ADCs as indicted in Table 2. Compared with a high-resolution pipelined ADC [30] and a high resolution SAR ADC [31], this work had a significant advantage of higher SNDR and SFDR due to the hybrid architecture of pipelined-SAR ADC with ISDM assisted. Besides, this kind of architecture could achieve a more power-efficiency merit compared to conventional pipeline ADC in high-precision field [32]. However, the total power consumption could be further modified because of the usage of GainBoost OTA in this work compared to [30] with a low power Ring AMP used.

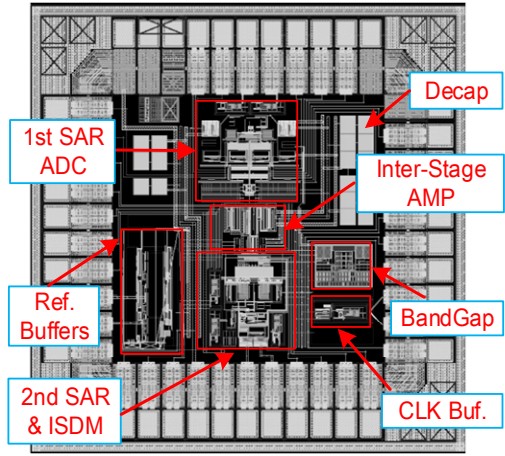

**Figure 17.** The layout photography of the proposed 16 bit pipelined-SAR ADC with ISDM assisted.

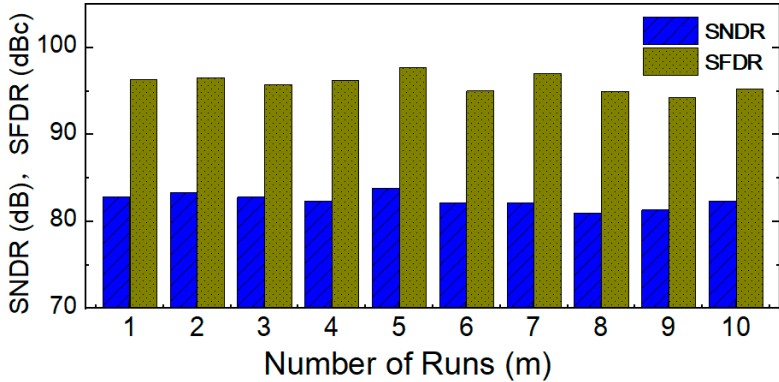

**Figure 18.** The SNDR and SFDR versus number of Monte-Carlo run under post-layout simulation with an input frequency of 2.3 MHz, supply ripples (1%) and transient noise.

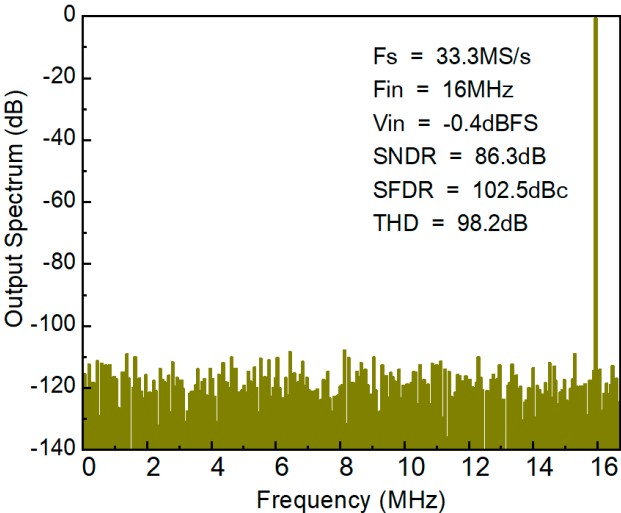

**Figure 19.** The output spectrum of the 16 bit pipelined-SAR ADC near Nyquist bandwidth under the post-layout simulation.

**Table 2.** The comparison with literature.

| Parameter | This Work [1] | [7] | [30] | [31] | [32] |
|---|---|---|---|---|---|
| Architecture | Pipe-SAR+ ISDM | Pipe | Pipe | SAR | Pipe |
| Amplifier | GainBoost | GainBoost | Ring-AMP | N.G. | ZCBC |
| Resolution (Bit) | 16 | 16 | 16 | 16 | 15 |
| Process (nm) | 40 | 180 | 90 | 55 | 130 |
| Area (mm²) | 0.202 | 9.9 | 0.368 | 0.55 | 0.96 |
| Supply (V) | 1.8/1.2 | 1.8 | 1.1 | 3.3/1.2 | 1.8/1.2 |
| Input Range (V) | 1.8 | N.G. | 1.1 | 6.6 | 2 |
| Power (mW) | 19.2 | 100 | 5.1 | 16.3 | 21.6 |
| Fs (MS/s) | 33.3 | 80 | 24 | 16 | 48 |
| Fin (MHz) | 16 | 9.7 | 10 | 0.1 | 5 |
| SFDR (dBc) | 102.5 | 95 | 89.4 | 98 | 95 |
| SNDR (dB) | 86.3 | 77.6 | 75.4 | 78 | 74.5 |
| FoMs (dB) | 175.5 | 163.6 | 169 | 165 | 165.1 |

[1] Post-Layout simulation results with transient noise.

## 5. Conclusions

In order to satisfy practical demands of high-resolution and moderate-bandwidth upon ADC, this paper proposed a kind of ISDM assisted 16-bit two-step pipelined SAR ADC architecture. The combination of two-stage low-resolution SAR ADC and low noise ISDM alleviated the resolution requirement of comparators in both stage of SAR ADC and achieved an improved SNDR about 11 dB without much hardware overhead. A designing example of the proposed ADC structure was also presented in a 40 nm CMOS process, simulation results of which show that this hybrid ADC could achieve the SDNR and SFDR no less than 86 dB and 102 dBc respectively while keeping a considerable figure of merits (FoMs) near 175 dB. Therefore, the proposed ADC architecture had an attractive and competitive application value in the future compared with those traditional ADC architectures.

**Author Contributions:** Conceptualization, B.G.; Formal analysis, X.L. and J.S.; Methodology, B.G. and X.L.; Project administration, J.W.; Visualization, X.L.; Writing—original draft, B.G.; Writing—review & editing, B.G., J.S. and J.W. All authors have read and agreed to the published version of the manuscript.

**Funding:** This research was funded by the Fundamental Research Funds for the Central Universities, grant number 2242018k30006 and the Natural Science Foundation of China, grant number 61871118.

**Conflicts of Interest:** The authors declare no conflict of interest.

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
