# Peer review of "Modeling of High-Resolution Data Converter: Two-Step Pipelined-SAR ADC based on ISDM"

_electronics, doi:10.3390/electronics9010137_

Round 1

Reviewer 1 Report

Design parameters and choices at the system level are not motivated, they are simply given as they are. Design choices may be illustrated through system level simulations, showing the trade offs. In my opinion this is relevant for the readers and will be useful for further development, since the architecture you propose is complex and aggregate various ADC techniques.

Author Response

Response to Reviewer 1 Comments

Dear Reviewer,

Thanks for your review about the paper named by “Modeling of High-Resolution Data Converter: Two-Step Pipelined-SAR ADC based on ISDM”. According to your practical and precious comments, I have modified the manuscript relatively and the point-to-point explanations or responses are given as follow.

Point1: Design parameters and choices at the system level are not motivated; they are simply given as they are. Design choices may be illustrated through system level simulations, showing the tradeoffs.

Response 1: Actually, Matlab model simulation of the whole ADC architecture has been finished before. While considering to the length of the pages, I deleted two simulation figures. Now, I will add them again as shown in figure 1, to illustrate the motivation and designing tradeoffs upon system level. More modified details can be found in section 2.1 The Proposed Architecture modeling of ADC.

Point2: In my opinion this is relevant for the readers and will be useful for further development, since the architecture you propose is complex and aggregate various ADC techniques.

Response 2: Indeed, comparing with single-stage SAR ADC and flash ADC as well as sigma-delta modulator, hybrid-structure ADCs, especially the pipelined ADCs, have more complex structure and higher design difficulty. However, we cannot ignore the merits that the hybrid ADCs can generate, and the two-step pipelined SAR ADC is a good example of them.

In the past few decades, researchers have been finding the most optimized ADC architectures to meet the sharply increasing demands of performance, such as high SNDR and SFDR as well as faster sampling rates while keeping better power efficiency. It is nearly impossible to achieve those requirements by using conversional methods in advanced CMOS process, as I mentioned in lines between 29 and 37.

In order to realize high performance, hybrid-structure ADCs come out to overcome the drawback or break the bottleneck of those traditional ADCs. Two-step pipelined SAR ADC makes good trade-off between resolution and power consumption as well as speed. But, it still meets challenging to achieve high SNDR (>80dB) among the-state-of-arts in recent years. In this paper, we want to deal with these problems by adding ISDM in the second stage SAR ADC. The benefits and rationality can be found in lines from 72 to 78 and section 2.1 The Proposed Architecture modeling of ADC. Considering to the hardware consumption, proposed structure only takes an extra open-loop integrator and a few digital logic circuits though, and the complexity of the whole ISDM-based pipelined-SAR ADC is not too high as well compared to the traditional pipelined ADC.

Last but not least, please allow us to show our thankfulness again for your revision on this paper.

Best regards,

Bo Gao.

Reviewer 2 Report

The objective of the article need to discussed clearly The application of the proposed SAR ADC should be added What is the purpose of the two-step pipelined SAR ADC? any evidence that results are improved because of pipelining. If you increasing the order of pipelining will further improve the results? Use proper scientific symbols, for example, inline 193, 199, 311 for microseconds or microvolts use µ than u GainBoost OTA, OTA is not abbreviated There are several devices that make similar ADC works from many companies like Analog Devices, Texas Instruments, Maxim, Intersil, STMicroelectronics, ON Semiconductor, Microchip, NXP Semiconductors, Cirrus Logic, Xilinx, Exar Corporation, ROHM Semiconductor, etc … why this work is essential? How to do perform calibration for Two-Step Pipelined-SAR irrespective of stability of power-supply voltages. Why offset-cancellation techniques are not used for comparators, which is common with many ADCs The integrator based comparator in SAR ADC needs to resolve voltages within the accuracy of the overall system. In the abstract and conclusion, it is mention that the proposed design as low noise high accuracy but didn’t provide sufficient proof. The conclusion should be improved

Author Response

Response to Reviewer 1 Comments

Dear Reviewer,

Thanks for your review about the paper named by Modeling of High-Resolution Data Converter: Two-Step Pipelined-SAR ADC based on ISDM. According to your practical and precious comments, I have modified the manuscript relatively and the point-to-point explanations or responses are given as follow.

Point 1: The objective of the article needs to be discussed clearly. 

Response 1: As you mentioned, the objective indeed needs to be addressed clearly. I modified the Introduction and first paragraph of section 1 to show more details of the motivation.

Point 2: The application of the proposed SAR ADC should be added. 

Response 2: Yes. I added some application areas in section 1, including in the medical imaging, radar and high precision industrial control, and etc. (see lines from 27 to 29)

Point 3: What is the purpose of the two-step pipelined SAR ADC? 

Response 3: The propose of the two-step pipelined SAR ADC is to realize high resolution and moderate sampling rate while keeping the resolution of each stage of the ADC no more than 10 bits and getting better SFDR by using the advantage of high linearity of SAR ADC in both stage, compared with traditional pipelined ADC. More importantly, this kind of structure decreases the designing difficulty or resolution of comparator used in first and second stage, which makes sure only the low-resolution kind of comparator needed in proposed ADC structure (see lines 71-78, 102-108).

Point 4: Any evidence is that results are improved because of pipelining. 

Response 4: The obvious improvement of pipelining is that the sampling rate has been increased significantly, which nearly doubles that of single-stage SAR ADC in high-resolution ADC categories so far. Furthermore, the pressure of comparator has been released remarkably compared with single-stage SAR ADC, where the resolution of comparator had to satisfy the whole resolution of ADC. In advanced CMOS process, this condition is quite challenging because of low intrinsic gain and low reference voltage (see lines 38-51).

Point 5: Whether you increasing the order of pipelining will further improve the results?

Response 5: It depends on the trade-offs between the resolution and power as well as complexity. The more pipelined stages added, the more resolution will be achieved in theory. However, for each additional stage of pipeline, one additional high performance gainboost OTA is needed, which means the power consumption will obviously increase (see lines 69-71). In addition, the linearity enhance techniques like injecting dither and more complex calibrations such as gain error calibration for each stage had to be added. Therefore, the two-step pipelined-SAR makes well compromises among those trade-offs.

Point 6: Use proper scientific symbols, for example, in line 193, 199, 311 for microseconds or microvolts use µ than u GainBoost OTA, OTA is not abbreviated 

Response 6: Yes, exactly. A few scientific symbols in this paper are not such formal. Thanks for your carefulness on this problem, and those symbols as you mentioned have been replaced in new submitted manuscript (See line 215, 221, 333).

Point 7: There are several devices that make similar ADC works from many companies like Analog Devices, Texas Instruments, Maxim, Intersil, STMicroelectronics, ON Semiconductor, Microchip, NXP Semiconductors, Cirrus Logic, Xilinx, Exar Corporation, ROHM Semiconductor, etc … why this work is essential? 

Response 7: Thanks for your examples of so many famous Inc. throughout the world. The ADC products from most of them focus on pipeline ADC for high-speed and high-resolution applications and SAR ADC or sigma-delta for low-power and low-speed fields, as those solutions are more mature. However, some specific application like military radar needs thousands of high-speed and high-resolution single channel ADCs combined to work together. Traditional structures like pipeline or SAR ADC cannot meet the power or speed requirement at the same time. This paper aims to overcome such bottleneck by using proposed ISDM-based pipelined-SAR ADC (see lines 69-78, 85-108).

Point 8: How do perform calibration for Two-Step Pipelined-SAR irrespective of stability of power-supply voltages. 

Response 8: A very good propose as you make. By post-layout simulation results we can find that the performance dropped when the ripple is added on the supply voltage. However, the ripple added is in an extreme situation and more dropped performance is caused by transient noise added in the simulation (see lines 396-398). In practical case, the well supply-voltage can be obtained by adding on-chip decouple capacitors and well-worked off-chip LDO. In addition, ADC works in a differential way, which means the ripple will have few inference on the reference voltage of DAC arrays. But, ripple will inference the working of gainboost OTA. Fortunately, we use band-gap to track the PVT variation and 1.8V supply voltage for OTA can make sure that there is enough overdrive voltage for each transistor near VDD or GND to immune the ripple. Therefore, the calibration seems not an essential part in this design.

Point 9: Why offset-cancellation techniques are not used for comparators, which is common with many ADCs 

Response 9: Actually, offset-cancellation is an indispensable part in the hybrid ADCs. In this paper, modified majoring-voting offset-calibration techniques has already existed in the first stage SAR ADC for comparator (see section 3.5 System Stabilization and Calibration), leaving the comparator used in second stage without offset-calibration, because offset voltage in second stage is constant and can be ignored when being equivalent to the input considering to the 32X gain of inter-stage gain amplifier.

Point 10: The integrator based comparator in SAR ADC needs to resolve voltages within the accuracy of the overall system.

Response 10: The design of integrator based comparator in second stage SAR ADC is very important as you mentioned, while the accuracy of it only need to meet the resolution demands of second stage SAR ADC (9 bits) actually, since the function of second ADC is to achieve total 11bit conversion (9 bits SAR conversion and 2bits ISDM conversion). Once this relationship has been determined, the only remained problem is how to make sure the correct work of ISDM-SAR ADC in second stage of overall system. For one thing, one bit over-range is added between SAR ADC and ISDM to make sure the quantization error of second stage SAR ADC is within the input range of ISDM. For another thing, the sharing comparator will cancel the offset between both of them.

Point 11: In the abstract and conclusion, it is mention that the proposed design as low noise high accuracy but didn’t provide sufficient proof. 

Response 11: Thanks for your advice about this problem. We have modified the abstract and conclusion. In fact, we want to express the opinion that proposed ADC can achieve moderate-bandwidth and well SNDR compared to traditional pipeline or SAR ADC (see lines 9-16, 413-416). 

Point 12: The conclusion should be improved 

Response 12: This is indeed lacking for well conclusion for this paper. We have modified the conclusion part (see lines from 412 to 420). 

Last but not least, please allow us to show our thankfulness again for your revision on this paper.

Best regards,

Bo Gao.
